# A Descriptive Study of Specialist and Non-Specialist Teachers’ Preparation towards Educational Inclusion

**DOI:** 10.3390/ijerph18147428

**Published:** 2021-07-12

**Authors:** Jorge Rojo-Ramos, Fernando Manzano-Redondo, Sabina Barrios-Fernandez, Miguel A. Garcia-Gordillo, Jose Carmelo Adsuar

**Affiliations:** 1Health, Economy, Motricity and Education (HEME) Research Group, Faculty of Sport Sciences, University of Extremadura, 10003 Cáceres, Spain; fmanzanoa@alumnos.unex.es (F.M.-R.); jadssal@unex.es (J.C.A.); 2Social Impact and Innovation in Health (InHEALTH) Research Group, Faculty of Sport Sciences, University of Extremadura, 10003 Cáceres, Spain; 3Faculty of Administration and Business, Universidad Autónoma de Chile, Sede Talca 3467987, Chile; miguel.garcia@uautonoma.cl

**Keywords:** inclusive education, teachers, initial preparation, therapeutic pedagogy

## Abstract

Attention to educational diversity in educational centers has become an important topic, so it is necessary to address challenges to offer an individualized educational response. Thus, specialist teachers must adopt a leading role in order that education systems move towards inclusion. The objective of this study is to measure Spanish primary school teachers’ perceptions about their preparation for inclusive education, considering possible differences between specialist teachers (therapeutic pedagogy and hearing and speech) and non-specialist teachers. The sample was made up of 284 teachers who work in the primary education stage in public Spanish schools, who responded to the Questionnaire for the Evaluation of Teacher Preparation for Inclusion (CEFI-R). Significant differences were found according to the specialism of the participants. It can be concluded that teachers consider their initial preparation in this subject insufficient but show positive conception towards educational inclusion.

## 1. Introduction

Inclusive education has become one of the major challenges for the educational systems, schools, teachers, and society [1]. Attention must be paid to diversity in order to face the growth of increasingly heterogeneous groups of pupils, and new responsibilities need to be incorporated into teaching tasks [2]. Educational institutions need to transform and adapt their educational programs to face this changing need. In this context, teachers must be prepared to deal with increasing diversity within their classrooms [3]. This social challenge may open up a new opportunity for the school to reinvent itself, and address necessary changes and adaptations to offer a higher quality educational response, adapted to their unique students [4].

The Department of Education and University Planning [5] defines educational attention to diversity as a set of ordinary and extraordinary measures which aim to adapt the educational response to the students’ characteristics, including their potential, learning rhythm and style, interests, social and cultural factors. Thus, teachers must design actions, adapt resources and the scholarly environment, rethinking the educational response and addressing individual differences [2]. Therefore, the opportunity for transformation and change towards attention to diversity cannot occur if quality education is not provided to all students [2], and teachers should be active in this process [6]. Muntaner [7] affirms that support models must be based on dialogue, reflection and the consequence of a climate of trust and joint work, actively participating in the educational culture, and being presented from a collaborative perspective, providing experience and specific preparation.

A key aspect of this process is teacher preparation [1]. Teacher preparation is essential in inclusive environments to reach higher quality education for every student. In this way, teacher preparation should be oriented towards the development of critical and actively collaborative educative professionals [1]. Subban and Mahlo’s study [8] showed how teachers who had undertaken inclusive education preparation showed better attitudes compared to those who had not received any specific preparation. The importance of teachers with expertise in therapeutic pedagogy and hearing and language should also be emphasized. These specialists have a complementary preparation that allows them to show a wider range of knowledge, resources, and strategies to collaborate with other teachers to help all the students to reach their educational goals [9], being instructed on the importance of multidisciplinary intervention [10]. Thus, specialist teachers’ functions include making individualized adaptations and monitoring them, organizing timetables, spaces and materials together with other members of the educational community, and working in the community and together with families, among others [9]. Thus, improving teachers’ attention to diversity competence is essential, developing their didactic skills, forms of organization and collaboration, and helping regular teachers to apply strategies and activities for educative inclusion [10]. 

In addition to the teachers’ initial preparation, other factors that influence the achievement of inclusive education are school leadership, collaborative culture, adaptations of equipment and infrastructure, multidisciplinary work, professional development, and teachers’ attitudes and perceptions [11]. The study carried out by Rojo-Ramos et al. [4] showed that future teachers’ attitudes toward inclusion are very positive, as they are motivated for change and aware of the difficulties and barriers. Teachers’ attitudes towards inclusive education is a key factor in supporting all students’ proper functioning in the classroom [12]. As a result, it has been shown that practicing primary school teachers tend to have more positive attitudes towards inclusive education than their counterparts in pre-primary and secondary education [8]. Various studies on attitudes have been conducted. Luque and Carrión [13] analyzed the functioning and underlying barriers for specialist teachers. Round et al. [14] explored teachers’ concerns about inclusive education in Australia and Saloviita [15] in Finland and Germany. Additionally, de la Rosa et al. [10] pointed out the need for better initial preparation in this area in specialist future teachers. Alba et al. [16] presented experiences perceived by teachers of therapeutic pedagogy with their students. De Boer’s literature review [17] examined what variables were related to inclusion and whether they affect students’ participation, as observed.

The Spanish education system is divided into several levels. The preschool stage is designed for children up to 6 years (non-compulsory). Primary education (compulsory) comprises six academic years between 6 and 12 years. Secondary education comprises Compulsory Secondary Education from the age of 12 to 16 (the last compulsory stage), and it can be followed by Professional Education or Baccalaureate. Regarding the initial training of primary school teachers, they obtain the specialist title when they complete a concrete pathway during their undergraduate preparation. Ongoing training is mainly supported through the Teachers’ and Resources Centers, which are devices managed by the education administration whose aim is to support their pedagogical work.

Most of the scientific literature regarding inclusive education has been produced in the last decade [18]. Thus, this study aims to measure Spanish primary school teachers’ perceptions about their preparation for inclusive education, and to find out whether there are differences according to the type of teacher, depending on whether they are specialist or expert teachers (therapeutic pedagogy and hearing and speech) or if they belong to the rest of the teaching staff.

## 2. Materials and Methods

### 2.1. Participants

The sample was made up of 284 teachers working in public schools at the primary education stage: 20.1% (57) men and 79.9% (227), women. Regarding their academic qualification or university degree, 44% (125) were specialists in educational inclusion (therapeutic pedagogy or hearing and speech), and 56% (159) were non-specialist. The mean number of years of experience of the teachers was 14.20 years (SD = 9.3). Participants were selected using a non-probability sampling method based on convenience sampling [19]. Table 1 shows the characterization of the participants.

### 2.2. Instruments

Sociodemographic data: Participants were asked about their sex, age, university degree and specialism, province of the school, teacher position, and years of teaching experience. Additionally, three questions about their preparation were asked: referring to (1) their initial preparation, (2) their ongoing preparation, and (3) their possible assistance to ongoing formation related to attention to diversity.

Teacher preparation for inclusion: The Questionnaire for the Evaluation of Teacher Preparation for Inclusion (CEFI-R) [20] was used (Appendix A). The CEFI-R is made up of 19 items grouped into four dimensions: (1) “Conception of diversity” (5 items) which measure beliefs of the concept of diversity, place, and form of schooling of students and educational policy on diversity; (2) “Methodology” (5 items), for aspects related to the design and development of an inclusive curriculum; (3) “Supports” (4 items), about teacher’s conception and role in this concept; and (4) “Community Participation” (5 items), which measures collaboration of all educational actors.

Before analyzing the results obtained, the indirect items were inverted. Thus, in these items, a higher score implies greater synchrony with an adequate “Conception of diversity” (dimension 1), greater consonance with an adequate “Methodology” (dimension 2), an adequate conception of “Support” (dimension 3) and a correct concordance with correct “Community participation” (dimension 4). This instrument uses a Likert scale where the values range from 1 to 4, 1 being “Strongly disagree”, 2 “Partially disagree”, 3 “Partially agree” and 4 “Strongly agree”.

The CEFI-R Cronbach’s alpha value is 0.79, and each of the four factors < 0.70, so it is a reliable instrument.

### 2.3. Procedure

The directory of the Ministry of Education and Employment of the Regional Government of Extremadura (Spain) was used to collect the emails from all the primary public schools in Extremadura. An email was sent providing information on the purpose of the research, written informed consent, and the URL to fill the questionnaires. All data were collected anonymously. The average time needed to answer the questionnaire was approximately 10 min. Data collection was performed using the Google Forms application, as electronic questionnaires have been proven to save costs and obtain higher participation [21]. The responses obtained were stored in a spreadsheet, facilitating their transformation and statistical analysis. Data collection was carried out between September and December 2020.

### 2.4. Statistical Analysis

Data were analyzed using the Statistical Package for Social Sciences (SPSS) version 23.0 for MAC (IBM Corporation, Armonk, NY, USA). The Kolmogorov Smirnov, asymmetry and kurtosis tests were performed to determine whether the data followed a normal distribution or not. This assumption was not met, so it was decided to use non-parametric tests.

The Pearson’s Chi-Square test was used to search for differences between the questions referring to initial and ongoing preparation according to the specialism of the teachers. The Mann–Whitney U test was used to analyze the relationships between the items of the CEFI-R dimensions according to the teachers’ specialism. The Spearman Rho was carried to measure the association between the CEFI four dimensions and the age groups. Cronbach’s Alpha was used to calculate the reliability of each of the dimensions in this study. Following the method of authors such as Nunnally and Bernstein [22], reliability values between 0.60–0.70 can be considered acceptable, while values between 0.70–0.90 can be considered satisfactory.

## 3. Results

### 3.1. The Three Questions Results

Differences between the questions referring to initial and ongoing preparation according to the specialism of the teachers were analyzed using Pearson’s Chi-Square test. Table 2 shows the frequency distribution of responses to the three items according to teachers’ specialism.

Distribution of frequencies to questions 1, 2 and 3 are shown in Table 2 and Figure 1, Figure 2 and Figure 3. In the first question, 31.7% (*n* = 90) answered yes, but 68.3% (*n* = 194) denied having been prepared to deal with the diversity of their students’ educational needs. The results showed that there was no significant association with the specialism (*p* = 0.05). Regarding the second question, 86.6% (*n* = 246) stated that they had been prepared through outgoing preparation to respond to the educational needs diversity, although 13.4% (*n* = 38) responded negatively to this aspect, and statistically significant differences according to the specialism were found (*p* < 0.01). Last, in the third question, 95.8% (*n* = 272) of the sample responded they would attend preparation courses in this respect, compared to 4.2% (*n* = 12) who responded negatively to this aspect. Again, statistically significant differences were found (*p* = 0.01).

### 3.2. The CEFI-R Questionnaire Results

Table 3 shows descriptive analysis and differences by specialism of the teachers in the items of the questionnaire.

Table 4 shows descriptive analysis and differences of each dimension of the CEFI-R questionnaire.

The first dimension “Conception of diversity” showed that Me = 3.4 and an IQR = 1. specialist teachers obtained slightly higher values than non-specialist teachers, with Me specialist = 3.6 (IQR = 0.8), Me non-specialist = 3 (IQR = 1.2), and statistically significant differences were found (*p* < 0.01). High scores (Me = 4; IQR = 1; specialism (*p* < 0.01)) were obtained in almost all the items, showing greater synchrony concerning an adequate conception of diversity.

The second dimension “Methodology” showed values of Me = 3; IQR = 1.2. Specialist teachers scored higher compared to non-specialists, with Me specialists = 3.6 (IQR = 1), and Me non-specialists = 3 (IQR = 1). Again, significant differences were obtained according to the specialism of the teachers (*p* < 0.01). The items belonging to this second dimension revealed similar values and significant differences depending on the specialism of the participating teachers (Me = 3; IQR = 1; specialism (*p* < 0.01)).

The third dimension, “Supports”, showed Me = 2.4 (IQR = 0.8). Likewise, specialist (Me specialists = 2.4 (IQR = 0.8)) and non-specialists (Me non-specialists = 2.4 (IQR = 1)) scores were similar. There were no significant differences (*p* = 0.13) according to the specialism. The highest score was obtained in item 11 (Me = 4, IQR = 1; specialism (*p* = 0.01)) and the lowest in item 13 (Me = 2, IQR = 2; specialism (*p* < 0.01)).

Concerning the fourth dimension, “Community Participation”, the scores showed Me = 3.8 (IQR = 0.8). Again, specialists showed higher values than non-specialists, with Me specialists = 4 (IQR = 0.4), Me non-specialists = 3.6 (IQR = 1), obtaining significant differences according to the teachers’ specialism (*p* < 0.01). All items showed similar scores, with a value of Me = 4 (IQR = 1), with statistically significant differences found in items 15 and 16 of these dimensions (*p* < 0.01).

### 3.3. Correlations between the CEFI-R Questionnaire and Age and Reliability

Finally, both Table 5 and Figure 4 show the correlations between CEFI-R dimensions and the age of the teachers using Spearman’s Rho test.

The results indicate that in dimension 1, “Conception of diversity”, there is a low and inversely significant correlation with age; as the age range increases, the score for dimension 1 decreases (ρ = −0.04; *p* = 0.61). The dimension “Methodology” showed a very similar correlation to the previous one, being low and inversely significant with age (ρ = −0.06; *p* = 0.44). Looking at the third dimension of “Supports”, it again showed a low and inversely significant correlation with age (ρ = −0.04; *p* = 0.58). Finally, the only dimension that shows statistically significant differences in its results is the fourth dimension “Community participation”, although it is still low (ρ = −0.20; *p* < 0.01).

Finally, the reliability results shows that the CEFI-R dimensions were: a ^1^ = 0.76; a ^2^ = 0.92; a ^3^ = 0.77 and a ^4^ = 0.92, with all values being satisfactory, all above 0.70 [22].

## 4. Discussion

This research was carried out to better understand preparation towards inclusive education in Spanish primary school teachers, describing differences depending on whether they were specialists or non-specialists. To achieve this goal, three questions and the CEFI-R questionnaire were used. 

Some studies [23,24,25] have highlighted the importance of the specialist role as a support for educative inclusion. In our study, participants answered three questions about initial preparation, ongoing preparation, and intention to attend preparation courses. Regarding the results obtained in question (1), whether they considered that their universities provided them with knowledge and tools to face the different needs of their students, there was a negative response of 68.3% from both specialist and non-specialist teachers. These results are not in line with those reported in the studies performed by Rojo-Ramos et al. [4], Sharma and Jacobs [26], or Rodríguez-Gómez et al. [27], which affirmed that inclusive education is effectively addressed in university degrees. In question (2) about ongoing preparation over the years, 86.6% of both specialist and non-specialist teachers attached great importance to lifelong learning courses. [8]. Lastly, in question (3), 95.8% of the participants showed their intention to attend preparation courses. [9]. Thus, to achieve inclusion, teachers need to receive initial and ongoing preparation which give them methodological tools to adapt to their students [28]. Thus, the results of questions (2) and (3) revealed statistically significant differences between specialist and non-specialist teachers.

In the CEFI-R dimension (1), “Conception of diversity”, all items showed high score values, and the item “Students with specific educational support needs can follow the day-to-day curriculum” was the one with the lowest value. In this respect, authors such as Benavides-Lara [29] or Duk and Loren [30] consider it necessary to review the curriculum and make adjustments to avoid inequalities. The results of this dimension are in line with other studies about educational diversity conception [4,26,31,32], suggesting that current and future teachers show positive attitudes towards educational inclusion. As González-Rojas and Triana-Fierro [33] pointed out, teachers play an important role in inclusion, and therefore their attitude is crucial to success in this process. In all the items included in this dimension, statistically significant differences were found according to the specialism of the participants. This may be in line with the findings of Luna and Martín [34], who indicate that specialist teachers tend to have more constructivist ideas and a more collaborative view of teaching and learning than non-specialist teachers.

In dimension (2), “Methodology”, the results indicate that both specialists and non-specialists felt competent to teach, adapt and develop for their students. Furthermore, in this dimension, statistically significant differences were found according to the specialism. In this regard, Orozco and Moriña [35] indicate that the methodologies and strategies used by the teachers were active, democratic and varied. Likewise, Majoko [28] highlights that specific methodological strategies can benefit all students. Gómez-Zepeda et al. [23] affirm that non-specialist teachers know their role and tried to work together with specialists. Therefore, it is important to provide the necessary educational means and create the most suitable conditions for the participation and progress of the entire educational community [35].

Dimension (3), “Supports”, was the one with the lowest values. The most positive values are found in the item “Joint teacher-support teacher planning would make it easier for support to be provided within the classroom”, highlighting that teachers approve of sharing knowledge regarding the needs of the group, guaranteeing more personalized attention [36]. The lowest value is found in the item “The role of the support teacher is to work with the whole class”. In this case, the dimension “Support” is the only one in which no statistically significant differences were found according to the specialism of the teachers.

Lastly, in dimension (4), “Community Participation”, statistically significant differences were found according to the teachers’ specialism. All the items addressed the urgent need for participation of all stakeholders involved in education to provide the best educational and inclusive response to the needs of students. [37,38]. Salinas [39] emphasized that, during the educational process, the cooperation with families and their involvement plays an important educational role, providing positive support for the students and providing valuable information.

In sum, the results on the four dimensions analyzed in the CEFI-R were quite positive. Statistically significant differences were found in dimensions 1, 2 and 4 when comparing specialist and non-specialist teachers. Schools must have teachers specialized in therapeutic pedagogy and in hearing and language to deal with the educational needs of every student [39]. However, González-Rojas and Triana-Fierro [33], and Rodríguez-Fuentes et al. [40] showed that sometimes, non-specialist teachers indicate that the responsibility for the care of students with special educational support needs lies with the specialist teacher, which can be a problem. Some studies [40] suggest that non-specialist teachers’ attitudes are acceptable, although improvement is possible, while specialist teachers are entirely positive.

Good reliability values are an indispensable characteristic for the use of these results [41]. In our study, values above 0.70 were obtained in all dimensions, in line with similar studies [4,20].

Given all the above, it seems clear that teachers must constantly update their preparation to ensure the quality of learning among diverse students. Specialized and properly trained teachers with the right attitudes and knowledge are required in order to respond properly to the needs of learners [42]. It is also vital that all those involved in the education system and society work together, to achieve the aim of true educational inclusion.

Our results should be interpreted considering several limitations. The sample was one of convenience. All participants were from the autonomous community of Extremadura, so socio-cultural factors could affect the results, which should be interpreted with care. Regarding future research, the recruitment of a larger sample, randomization of participants and multi-center studies would be interesting alternatives to explore.

## 5. Conclusions

Both specialist and non-specialist teachers consider that their preparation concerning inclusive education should be improved, with greater emphasis on ongoing preparation. Participants demonstrate positive attitudes towards inclusion, recognizing the importance of multidisciplinary work, feeling competent to teach and being able to develop appropriate teaching materials adapted to their students. Statistically significant differences were found between specialist and non-specialist teachers in the application of the CEFI-R in three of the dimensions: (1) “Conception of diversity”, (2) “Methodology”, and (4) “Community Participation”.

## Figures and Tables

**Figure 1 ijerph-18-07428-f001:**
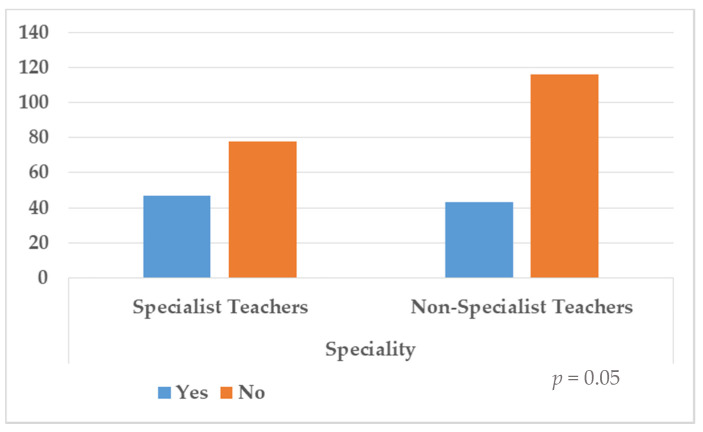
Distribution of responses according to question 1.

**Figure 2 ijerph-18-07428-f002:**
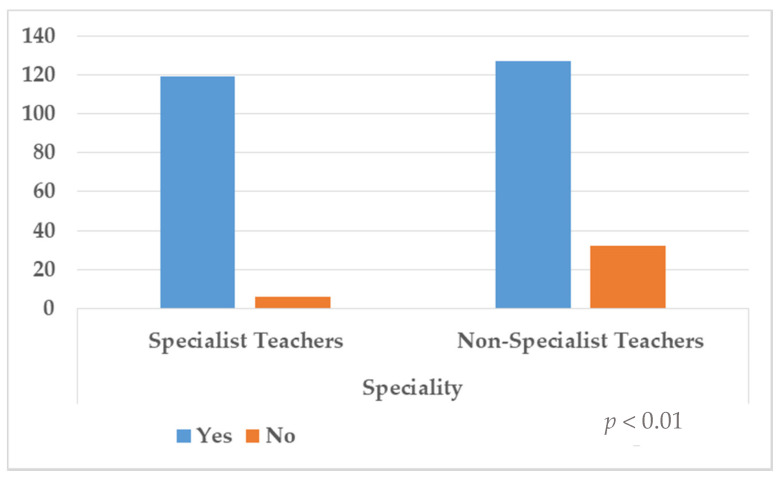
Distribution of responses according to question 2.

**Figure 3 ijerph-18-07428-f003:**
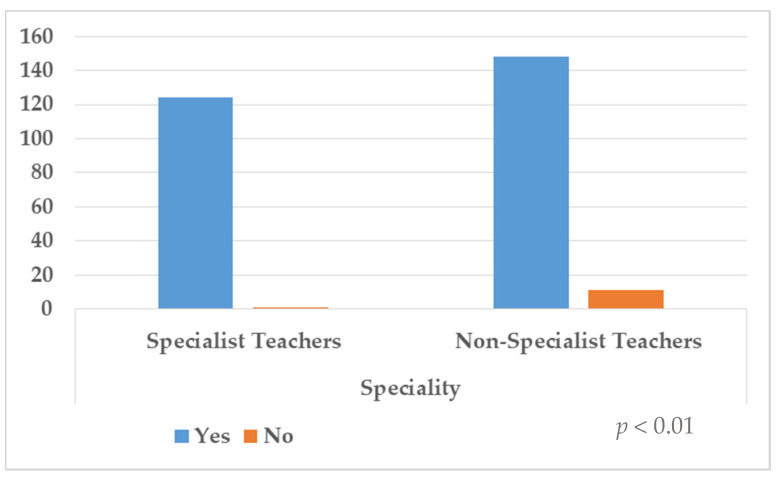
Distribution of responses according to question 3.

**Figure 4 ijerph-18-07428-f004:**
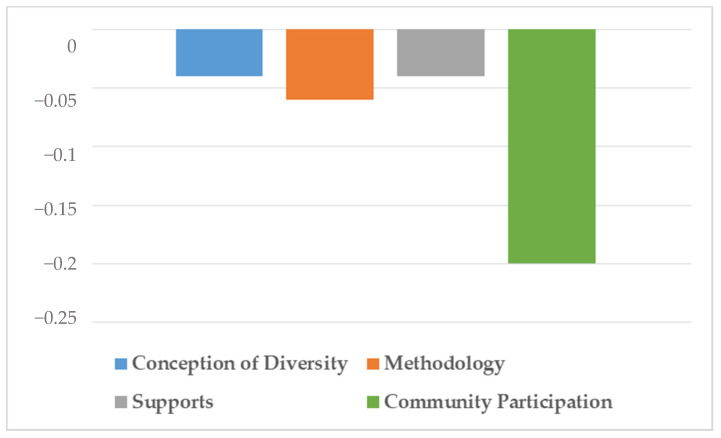
Differences in correlations between the dimensions and the age group variable.

**Table 1 ijerph-18-07428-t001:** Characteristics of the sample (*N* = 284).

Variable	Categories	*N*	%
Sex	Men	57	20.1
	Women	227	79.9
Age	Under 30	19	6.7
	Between 30 and 40	116	40.8
	Between 41 and 50	98	34.5
	Over 50	51	18
University degree and specialism	Primary Education (Non-specialists)	159	56
	Primary Education (Therapeutic Pedagogy)	79	27.8
	Primary Education (Hearing and Speech)	46	16.2
Province of the school	Cáceres	65	22.9
	Badajoz	219	77.1
Teacher position	Contract teacher position	89	31.3
	Permanent teacher position	195	68.7
Years of experience		14.20 (mean)	9.30 (sd)

**Table 2 ijerph-18-07428-t002:** Distribution of responses to the questions according to the specialism of the teachers.

*“My initial preparation prepared me to respond to the diversity diversity of my students needs”*					
			*Yes*	*No*	*p*-Value
Specialism	Specialist teachers	*N*	47	78	0.0
		%	52.2	40.2	
	Non-Specialist teachers	*N*	43	116	
		%	47.8	59.8	
	Total	*N*	90	194	
		%	31.7	68.3	
*“Ongoing preparation over the years has prepared me to respond to the diversity of my students needs”*					
			*Yes*	*No*	*p*-Value
Specialism	Specialist teachers	*N*	119	6	<0.01
		%	48.4	15.8	
	Non-Specialist teachers	*N*	127	32	
		%	51.6	84.2	
	Total	*N*	246	38	
		%	86.6	13.4	
*“If I were offered inclusive preparation in this respect, I would attend.”*					
			*Yes*	*No*	*p*-Value
Specialism	Specialist teachers	*N*	124	1	0.01
		%	45.6	8.3	
	Non-Specialist teachers	*N*	148	11	
		%	54.4	91.7	
	Total	*N*	272	12	
		%	95.8	4.2	

Pearson’s Chi-Square *p*-value test.

**Table 3 ijerph-18-07428-t003:** Descriptive analysis and differences by specialism of the teachers in the items of the questionnaire.

		Specialism		
Item	Total	Specialist Teachers	Non-Specialist Teachers	
	M_e_ (IQR)	M_e_ (IQR)	M_e_ (IQR)	*p*-Value
I would prefer to have students with specific educational support needs in my classroom.	4 (1)	4 (0)	4 (1)	<0.01
2. A child with specific educational support needs does not disrupt the classroom routine and disrupt the learning of his/her classmates.	4 (1)	4 (1)	3 (2)	<0.01
3. We should place students with special educational needs in mainstream schools even if we do not have the appropriate preparation to do so.	4 (1)	4 (1)	3 (2)	<0.01
4. Students with specific educational support needs can follow the day-to-day curriculum.	3 (1)	3 (1)	3 (2)	<0.01
5. I am not worried that my workload will increase if I have students with specific educational support needs in my class.	4 (1)	4 (1)	3 (2)	<0.01
6. I know how to teach each of my students differently according to their characteristics.	3 (2)	4 (1)	3 (1)	<0.01
7. I know how to design teaching units and lessons with the diversity of students in mind.	3 (2)	4 (1)	3 (2)	<0.01
8. I know how to adapt the way I assess the individual needs of each of my students.	3 (1)	4 (1)	3 (2)	<0.01
9. I know how to handle and adapt teaching materials to respond to the needs of each of my students.	3 (1)	4 (1)	3 (2)	<0.01
10. I can adapt my communication techniques to ensure that all students can be successfully included in the mainstream classroom.	3 (1)	4 (1)	3 (2)	<0.01
11. Joint teacher-support teacher planning would make it easier for support to be provided within the classroom.	4 (1)	4 (1)	4 (1)	0.01
12. I believe that the best way to provide support for students is for the support teacher to be embedded in the classroom, rather than in the support classroom.	3 (2)	3 (2)	3 (2)	0.38
13. The role of the support teacher is to work with the whole class.	2 (2)	2 (1)	2 (2)	<0.01
14. I consider that the place of the support teacher is in the regular classroom with each of the teachers.	3 (2)	3 (2)	3 (2)	0.80
15. The educational project should be reviewed with the participation of the different agents of the educational community (teachers, parents, students).	4 (1)	4 (1)	3 (2)	<0.01
16. There must be a very close relationship between the teaching staff and the rest of the educational agents (AMPA, neighbourhood associations, school council, …).	4 (1)	4 (1)	4 (1)	<0.01
17. The school must encourage the involvement of parents and the community.	4 (1)	4 (1)	4 (1)	0.16
18. Each member of the school (teachers, parents, students, other professionals) is a fundamental element of the school.	4 (0)	4 (0)	4 (1)	0.13
19. The school must work together with the resources of the neighbourhood (library, social services, health services, etc.).	4 (1)	4 (1)	4 (1)	0.12

Note: Me = median value; IQR = interquartile range. Each score obtained is based on a Likert scale (1–4): (1) being “Strongly Disagree”, (2) “Partially Disagree”, (3) “Partially Agree” and (4) “Strongly Agree”. Mann–Whitney U test *p*-values.

**Table 4 ijerph-18-07428-t004:** Descriptive analysis and differences of each dimension of the questionnaire.

		Specialism		
Dimensions	Total Me (IQR)	Specialist Teachers	Non-Specialist Teachers	*p*
Conception of Diversity	3.4 (1)	3.6 (0.8)	3 (1.2)	<0.01
2. Methodology	3 (1.2)	3.6 (1)	3 (1)	<0.01
3. Supports	2.4 (0.8)	2.4 (0.8)	2.4 (1)	0.13
4. Community Participation	3.8 (0.8)	4 (0.4)	3.6 (1)	<0.01

Note: Me = median value; IQR = interquartile range. Each score obtained is based on a Likert scale (1–4): (1) being “Strongly Disagree”, (2) “Partially Disagree”, (3) “Partially Agree” and (4) “Strongly Agree”. *p*-values from the Spearman Rho test.

**Table 5 ijerph-18-07428-t005:** Correlations between the CEFI-R dimensions and the variable age groups.

Dimensions	Age ρ (*p*)
Conception of diversity	−0.04 (0.61)
Methodology	−0.06 (0.44)
Supports	−0.04 (0.58)
Community Participation	−0.20 (<0.01)

Each score was obtained on the dimensions Likert scale (1–4): 1 being “Strongly Disagree”, (2) “Partially Disagree”, (3) “Partially Agree” and (4) “Strongly Agree”.

## Data Availability

The datasets used during the current study are available from the corresponding author on reasonable request.

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
