# Peer review of "A Descriptive Study of Specialist and Non-Specialist Teachers’ Preparation towards Educational Inclusion"

_ijerph, 2021, doi:10.3390/ijerph18147428_

Round 1

Reviewer 1 Report

I enjoyed reading this article and it adds some important research to the field of teacher preparation for inclusive education. The rationale for the paper is clearly articulated in the introduction and the integration of wide ranging empirical literature adds some context.  Real strengths of the paper include the methodology section which is clear and transparent. Statistical analysis is clearly defined, although for readers not familiar with inferential statistics and quantitative analysis it might be worth considering the inclusion of some brief narrative after each table summarising the data. 

As a reviewer not entirely familiar with the Spanish system of teacher education I have a couple of recommendations which may be helpful to an international audience:

  • In the literature review/ introduction I think it would benefit from inclusion of a discrete section which describes how teacher education is delivered in Spain. For example, can pre-service teachers qualify as specialist teachers at undergraduate level? How does the specialist teacher become qualified? Who provides in-service professional development or CPD in the area of inclusive and special education? Does the Ministry of Education support this or is it privately operated? How are teachers in Spain supported to engage in professional development? Is it individual or whole-school? 
  • Use of the word teacher 'training' may be problematic from an international perspective. Many jurisdictions now move to the language of teacher preparation/ teacher education at initial teacher education level and continuing professional development (CPD) or teacher professional learning (TPL) at the in-service stage of the continuum of teacher education. The type of teacher education the authors aim for and describe in this paper appear to be embedded in the conceptual understanding of teacher professional development rather than 'training' and emphasise the importance of collaborative learning for inclusive education. I would recommend the authors take a look at some of the reports which have been published by the European Agency for Special Needs and Inclusive Education (EASNIE) on 'Teacher Professional Learning for Inclusion (TPL4I). 

In relation to accessibility of the language, for the most part it was very good. The initial literature review section, while accessible, might benefit from some minor rephrasing of some sentences which read like a direct translation from Spanish. 

I spotted a typo on page 3, line 128- it says 'teachers' but it should be 'teacher' I think. 

Finally, this is a solid paper which adds to a well developed field of research on teacher education for inclusive education and the conclusions and recommendations will be helpful to other countries. The potential for some collaborative and comparative research across nations could be facilitated by way of the excellent survey, which seems entirely robust and empirically validated. It would be helpful if the survey could be added to the supplementary materials with permission. 

Well done to the authors. It is a well constructed paper and I have no hesitation in recommending it for publication. 

Reviewer 2 Report

First of all, I would like to congratulate the authors for their 
their article on a descriptive study of the training and attitudes of specialist
and attitudes of specialist teachers towards inclusive education.
towards educational inclusion.

The abstract is correct and provides sufficient information about the article.

The introduction has important theoretical references from the last five the last five years, so it is up to date and well constructed.

In relation to the empirical framework, the sample is well defined; regarding the instrument, an already constructed instrument has been used. This is always a problem, since each investigation has its peculiarities
This is always a problem, since each investigation has its peculiarities and sometimes it is better to construct the instrument itself for the investigation, for this reason, it would be convenient to talk about the content validity of the instrument, since the reliability is not very good, at least the content can be.

Subsequently, a frequency analysis is performed, they have used the median, right. 
used the median, correct. It does not clarify the relationship
between the dimensions and the group.

Since it was not possible to perform an anova, the Mann Whitney U test was used, this could be explained a little more. 

It would have been interesting, in this descriptive analysis, to analyze the asymmetry and curvature of skewness and kurtosis to better clarify the data distribution.

Otherwise, the conclusions are correct. My congratulations.
